# Identifying the NEAT1/miR-26b-5p/S100A2 axis as a regulator in Parkinson's disease based on the ferroptosis-related genes

Taole Li[1], Jifeng Guo[1,2,3,4,5]*

1 Department of Neurology, Xiangya Hospital, Central South University, Changsha, China, 2 Hunan Key Laboratory of Medical Genetics, School of Life Sciences, Centre for Medical Genetics, Central South University, Changsha, China, 3 Key Laboratory of Hunan Province in Neurodegenerative Disorders, Central South University, Changsha, China, 4 National Clinical Research Centre for Geriatric Disorders, Xiangya Hospital, Central South University, Changsha, China, 5 Hunan International Scientific and Technological Cooperation Base of Neurodegenerative and Neurogenetic Diseases, Changsha, China

* guojifeng@csu.edu.cn

## Abstract

### Objectives

Parkinson's disease (PD) is a complex neurodegenerative disease with unclear pathogenesis. Some recent studies have shown that there is a close relationship between PD and ferroptosis. We aimed to identify the ferroptosis-related genes (FRGs) and construct competing endogenous RNA (ceRNA) networks to further assess the pathogenesis of PD.

### Methods

Expression of 97 substantia nigra (SN) samples were obtained and intersected with FRGs. Bioinformatics analysis, including the gene set enrichment analysis (GSEA), consensus cluster analysis, weight gene co-expression network analysis (WGCNA), and machine learning algorithms, were employed to assess the feasible differentially expressed genes (DEGs). Characteristic signature genes were used to create novel diagnostic models and construct competing endogenous RNA (ceRNA) regulatory network for PD, which were further verified by in vitro experiments and single-cell RNA sequencing (scRNA-seq).

### Results

A total of 453 DEGs were identified and 11 FRGs were selected. We sorted the entire PD cohort into two subtypes based on the FRGs and obtained 67 hub genes. According to the five machine algorithms, 4 features (*S100A2*, *GNGT1*, *NEUROD4*, *FCN2*) were screened and used to create a PD diagnostic model. Corresponding miRNAs and lncRNAs were predicted to construct a ceRNA network. The scRNA-seq and experimental results showed that the signature model had a certain diagnostic effect and lncRNA *NEAT1* might regulate the progression of ferroptosis in PD via the NEAT1/miR-26b-5p/S100A2 axis.

**Data Availability Statement:** All datasets can be downloaded in the GEO database (http://www.ncbi.nlm.nih.gov/geo). The accession numbers: GSE7621, GSE49036, GSE26927, GSE20292,

GSE114517 and GSE178265. The ferroptosis genes can be downloaded in the FerrDb database (web URL http://www.zhounan.org/ferrdb/). Further inquiries can be directed to 198102101@csu.edu.cn.

**Funding:** This research was funded by the National Natural Science Foundation of China (Grant No. 81873785) and the Technology Major Project of Hunan Provincial Science and Technology Department (Grant No. 2021SK1010). The funders had no role in study design, data collection and analysis, decision to publish, or preparation of the manuscript.

**Competing interests:** The authors declare that there are no competing interests associated with the manuscript.

## Conclusion

The diagnostic signatures based on the four FRGs had certain diagnostic and individual effects. NEAT1/miR-26b-5p/S100A2 axis is associated with ferroptosis in the pathogenesis of PD. Our findings provide new solutions for treating PD.

## 1.Introduction

PD is the second most common progressive neurodegenerative movement disorder, characterized by bradykinesia, postural instability, muscle rigidity, resting tremors, sleep, and thinking problems [1]. Its main neuropathological changes include the loss of dopaminergic neurons in the SN pars compacta and misfolded a-synuclein in Lewy bodies. Current drug therapy for PD is symptomatic and primarily relies on the restoration of dopaminergic function in the striatum. However, long-term use of dopamine drugs has been associated with disabling complications, including dyskinesias and fluctuating motor responses. Also, considering that some other neurological conditions have similar clinical features, there is still uncertainty regarding the diagnosis, particularly in the early stages [1]. Thus, reliable diagnostic or prognostic biomarkers are urgently needed for disease management.

Studies have revealed that metal elements, such as manganese, copper, and iron, have a significant role in PD [2]. These metals can intricately impair several aspects of neurological functions, such as oxidative stress, mitochondrial or lysosomal dysfunction, and synaptic inflammation [3], and are also involved in multiple mechanisms of cell death patterns such as autophagy, pyroptosis, ferroptosis, and cuproptosis [4,5]. Ferroptosis is a newly-regulated cell death mode caused by the abnormal increase in iron-dependent lipid reactive oxygen species and the imbalance of redox homeostasis [6]. Ferroptosis is regulated by a number of cellular variables, including iron metabolism, lipid incorporation, biosynthesis of glutathione (GSH), glutathione peroxidase 4 (GPX4), NADPH, and CoQH2 [7]. Ferroptosis is highly negatively correlated with cancer development but positively correlated with some neurodegenerative diseases, including PD. The traits of ferroptosis induction, such as brain iron overload, PUFA-PLs producing reactive oxygen species (ROS), and the exhaustion of the xCT system, are remarkably compatible with the PD models [8]. Moreover, previous studies discovered that PD-deleterious genes (e.g., *Parkin*, *LRRK2*, *DJ-1*, *PLA2G6*) are associated with ferroptosis [9–11]. The α-synuclein, also as an iron-binding protein, could modulate the levels of iron transport or lipid metabolism [12]. Some preliminary data have shown that ferroptosis inhibitors could prevent ferroptosis and limit neurodegeneration in PD, but also amyotrophic lateral sclerosis (ALS), Alzheimer's, and other diseases with brain iron dysregulation. Thus, genetic testing could be a powerful tool to uncover biological pathways that cause PD.

In this study, we investigated the pathogenesis of PD by searching for ferroptosis-related genes (FRGs), which were then used to construct ceRNA network using bioinformatics analysis combined with experimental validation. Gene Expression Omnibus (GEO) [13] database intersected with the ferroptosis dataset (FerrDb) was used to confirm the expression of FRGs. Then, we investigated the consensus clustering analysis and constructed the co-expression network. We also performed GSEA, WGCNA, biological function analysis of Gene Ontology (GO), and Kyoto Encyclopedia of Genes and Genomes (KEGG) to explore the crucial signature genes linked with PD in hub modules. Furthermore, the diagnostic classifiers of PD related to ferroptosis based on five machine-learning algorithms were constructed. Hub features were selected, and their diagnostic value as the biomarkers or predictive model of PD

was assessed. Finally, targeted miRNA and lncRNA were predicted by the ceRNA network. The scRNA-seq and experimental validation confirmed the role of the features. In light of our study, lncRNA *NEAT1* was regarded as a responsible biological regulator of ferroptosis in the onset and progression of PD.

## 2. Materials and methods

### 2.1. Data collection and processing

RNA sequencing data from PD patients, including microarray and next-generation sequencing (NGS), were obtained from the GEO database. Four microarray profile data, i.e., GSE7621, GSE49036, GSE26927, and GSE20292, containing 97 post-mortem SN samples originating from PD donors were downloaded. The "Affymetrix" datasets were background corrected or normalized by using the R package "affy" within "Robust Multiarray Analysis" algorithm. The "Illumina" datasets were background corrected or normalized by using the R package "limma" within "read.ilmn" or "neqc"algorithm. The matching of probes and genes was achieved by using the R package "AnnotationDbi" and "org.Hs.eg.db", especially the R package "hugene-e10sttranscriptcluster.db" were used to filter out the probes without corresponding genes. The log2 transformation and normalization were conducted by using the R package "geoquery" within "gds2eset" algorithm. By optimizing R package "sva", We constructed the model using batch information as a covariate and generated the expression matrix, including 54 PD samples and 43 normal control (NC) samples. NGS data of GSE114517, including 46 PD samples and 29 normal samples, were extracted for confirmatory studies. The quality control were performed by using the R package "arrayQualityMetrics" (**S1 Fig**). Also, 564 FRGs related to ferroptosis signal pathways were obtained from the FerrDB database. These datasets were employed for further analysis and mining.

### 2.2. DEGs analysis

The DEGs between PD patients and NC samples were identified using the R-package "limma". The confounding factors were addressed within "lmfit" function and "Bayesian" testing.

Considering that subtle genetic differences in neurodegenerative diseases such as PD may lead to the significant changes in molecular biological mechanisms. The following criteria were applied: thresholds at the P value$<0.05$ (p-value $< 0.05$) and the absolute log2 fold change (log2FC) $> 0.1$ (log2|FC| $> 0.1$). A total of 564 FRGs were intersected after merging four microarray datasets to identify the FRGs. Expressions of DEGs were presented as heatmaps or volcano plots by using R package "ggplot2" and "pheatmap".

### 2.3. Consensus clustering

Consensus clustering is a useful algorithm for identifying distinct ferroptosis-related patterns, which rely on k-means analysis. In this study, we selected FRGs of DEGs in PD patients for further analysis. Based on the expression of FRGs packed above, the R package "ConsensusClusterPlus" was employed for consensus unsupervised clustering to divide PD patients into distinct molecular subtypes according to FRGs expression. Cumulative distribution function (CDF) was applied to choose the optimal cluster number.

### 2.4. GSEA

"GSEA" software was used to investigate involved GO-KEGG pathways of the reference DEGs between PD vs controls or two clusters. NOM p-value $< 0.05$ was defined as the significant enrichment.

## 2.5. WGCNA

WGCNA was used to assess the relationships between hub gene modules and clinical traits of PD. To ensure the accuracy of identified the hub modules significantly related to clinical features of PD, the age-matched PD and NC samples were singled out from GSE20292, including 11 PD patients and 11 NC samples. The dataset has well-documented clinical information with appropriate number and high quality of samples. The co-expressed module containing parallel expression patterns was constructed. Based on the R function of "PickSoftThreshold" algorithm, an appropriate soft threshold power was selected. Then, the dynamic tree cut function or hierarchical clustering was performed to divide the different modules from all genes, and similar models were incorporated using MEDissThres = 0.4. Subsequently, Module Membership (MM) combined with Gene significance (GS) was defined as indicatrix for the chosen genes that originate from the module eigengenes.

## 2.6. Functional and pathways enrichment analysis

To explore the possible molecular functions of hub genes associated with PD, GO, and KEGG enrichment analysis was applied to the PD-related modules by the WGCNA analysis. The R package "Enrichplot" and "ClusterProfiler" was employed, and the enrichment of significance was adjusted to P-value < 0.05.

## 2.7. Machine learning algorithms

Hub genes were slected to constitute the features by using five machine learning algorithms, including LASSO regression, Random Forest (RF), eXtreme Gradient Boosting (XGBoost), Gradient Boosting Machines (GBM) and Support Vector Machines (SVM). A total of 97 samples from four datasets were randomly separated into train and test sets at a 7:3 ratio by functional createfolds with R package "caret". GSE114517 was set as an external validation dataset. The R package "glmnet" was used to perform the LASSO algorithm with parameters set as set. seed (1) and family = "binomial". Adding a penalty term and the L1 penalty was selected based on the cross-validation results to reduce model overfitting. The R package "randomForest" was utilized to conduct the RF algorithm. The criterion for feature importance was MeanDecreaseGini index > 1.5. The R packages "xgboost" and "gbm" were respectively used for XGBoost and GBM algorithm. The model performance was optimized by constructing decision trees incrementally and the core features were selected according to the rank of their importance scores. The R package "e1071" was employed to SVM and provide utilities for model training and classification operations. The number of feature genes were selected and determined with the highest accuracy and lowest error rate. The 10-fold or 5-fold cross-validation was utilized to optimize the hyperparameters of machine learning model. After investigating the intersection point of five machine algorithms learning, the PD diagnostic model was formulated with the remaining features.

## 2.8. Construction and validation of classifier model

The intersecting genes of five machine learning algorithms were used to construct an PD diagnostic model through multivariable logistic regression analysis based on the "rms" R package. The PD diagnostic scores were performed according to the following formula:

$$\text{Diagnostic Model} = \sum_{i=1-4}^{Expi} Expi * coefi$$

where Exp stands for standardized gene expression, i is the number of diagnostic genes, and coef represents regression coefficients.

The diagnostic score was calculated for each patient and the receiver operating characteristic curve (ROC) was drawn to evaluate the predictive accuracy of the signature via the "pROC" R package. The area under the ROC curve (AUC) was used to assess the degree of sensitivity and specificity of the diagnostic model. The AUC > 0.7 was considered indicative of high diagnostic performance.

## 2.9. Prediction of miRNA–lncRNA and construction of ceRNA network

The targeted pivotal miRNAs of hub signature genes were predicted using the "NetworkAnalyst", which is based on the "miRTarBase" "miRWalk" and "miRDB" databases. In addition, the miRNAs-related lncRNAs were selected by using the StarBase v2.0 with a high stringency confidence level (degree > 5). The ceRNA network of cross-linked lncRNA–miRNA–mRNA was constructed and visualized by Cytoscape v3.9.1.

## 2.10. Cell cultures and drug treatment

Human neuroblastoma SH-SY5Y cells (#CL-0208, Procell, Wuhan, China) were cultured in a specific medium (#CM-0208, Procell, Wuhan, China) in a humidified atmosphere containing 5%$CO_2$/95% air at 37°C. Cells were seeded in 1.5ml medium at a density of $10^5$ per well in a 12-well plate for 1 day. Then, cells were respectively treated with indicated drugs. Ctrl group, cells were treated with 0.1% dimethyl sulfoxide (#D8418, Sigma Aldrich, Louis, MO, USA). Erastin group, cells were treated with 10μM Erastin (#S7242, Selleck, Houston, USA). 6-OHDA group, cells were treated with 50μM 6- hydroxydopamine (6-OHDA) (#S5324, Selleck, Houston, USA) (dissolved in 0.1% dimethyl sulfoxide). 6-OHDA+Lip-1 group, cells were treated with 50μM 6-OHDA and 5 μM liproxstatin-1 (Lip-1) (#S7699, Selleck, Houston, USA). For the transfection, cells were mixed with si-RNA (si-NC, si-NEAT1) or miRNA inhibitor (NC-inhibitor, 26b-5p inhibitor) (Genepharma, Suzhou, China) and lipofectamine 3000 (#L3000001, Invitrogen, Carlsbad, USA) dissolved in 100 μL Opti-MEM (#11058021, GIBCO, Grand Island, USA) for 15 minutes. After the transfection, the SH-SY5Y cells were cultured with standard growth medium for 8-12h and then treated with 50μM 6-OHDA. After 36h treatment, the cells were collected and subjected to experiments.

## 2.11. Western blot

SH-SY5Y cells were mixed with RIPA lysis buffer (#YSD0100, Yoche, Shanghai, China). Equal protein was separated using 4%-12% sodium dodecyl sulfate-polyacrylamide gels and transferred to polyvinylidene fluoride membranes by electroblotting. After being sealed with 5% milk for 1h, the membranes were incubated with primary antibodies tyrosine hydroxylase (TH) (#sc25269, Santa Cruz Biotechnology, Dallas, USA), acyl-CoA synthetase long-chain family member 4(ACSL4) (#P07940, Promab, Changsha, China), GPX4 (#ab125066, Abcam, Cambridge, USA), transferrin receptor(TFR) (#ab84036, Abcam, Cambridge, USA), ferritin heavy chain 1 (FTH1) (#4393, Cell Signaling Technology, Danvers, USA), ACTIN (#81115-1-RR, Proteintech, Chicago, USA) overnight at 4°C. The HRP-conjugated secondary antibodies were incubated at room temperature for 1h. Finally, membranes were visualized using the ECL reagents and quantified using Image J software.

## 2.12. Measurement of iron, lipid peroxidation, and GSH

SH-SY5Y cells were collected with centrifugation at 800 rpm at 4°C for 5 min. The quantifications of iron (#ab83366, Abcam, Cambridge, USA), lipid peroxidation (MDA) (#A003-1, Jiancheng Bioengineering Insitute, Nanjing, China), and GSH (#A006-1-1, Jiancheng

Bioengineering Insitute, Nanjing, China) were then performed, following the manufacturer's instructions.

## 2.13. RNA extraction and quantitative real Time-PCR (qRT-PCR) analysis

Total RNA was extracted from SH-SY5Y cells using the TRIzol and chloroform reagent (Invitrogen, Carlsbad, USA) in frozen environment. The cells were homogenised in 500 μL Trizol and 100 μL chloroform incubated at room temperature for 5 min, then centrifuged at 13000×g for 15 min at 4˚C. The aqueous phase containing RNA was extracted and added equal volumes of isopropanol (Sigma-Aldrich, Louis, MO, USA). The samples were centrifuged at 13000×g for 15 min at 4˚C and washed with ethanol. The RNA was air dried and dissolved in nuclease-free water, then the purity and quality were detected by using spectrophotometer. The genomic DNA was removed and cDNA was obtained using a reverse transcription PCR kit (#R223/ MR201, Vazyme, Nanjing, China) according to the manufacturer's instructions. 1 μg of RNA was used to synthesize cDNA in 20 μl reverse transcription reaction, followed by 50˚C for 15 min and 85˚C for 5 s. The qRT-PCR was achieved by using the ChamQ Universal SYBR qPCR Master Mix (#Q711, Vazyme, Nanjing, China) and miRNA Universal SYBR qPCR Master Mix (#MQ101, Vazyme, Nanjing, China) as per the manufacturer's protocol. Primer sequences are listed in **S1 Table**. The length of the amplification product is between 80 and 200bp with a Tm of 60˚C. The amplification efficiency ≥ 90% (1.8–2.2) tested by the standard curve, and a single melt curve, as a standard for qualification. The qRT-PCR were performed in 20 μl reaction on ABI 7300 Real Time PCR system, followed by 95˚C for 30s, 40 cycles of 95˚C for 10 s and 60˚C for 30 s, 95˚C for 15 s, 60˚C for 60 s and 95˚C for 15 s. All measurements were performed in triplicate. The relative change of mRNA, miRNA, and lncRNA levels were measured by 2-△△Ct method and the Ct values were normalized to GAPDH or U6.

## 2.14. Measurement of ROS

The level of ROS in SH-SY5Y cells was assessed using a ROS assay kit (#S0033S, Beyotime, Shanghai, China). Briefly, cells were stained with 10 μM of DCFH-DA working solution (1:1000) and then incubated for 20–30 min at 37˚C in the dark. Next, cells were washed three times with PBS buffer to clear the excess probe. Finally, the fluorescences of DCFH-DA were visualized under the inverted fluorescence microscope with an excitation wavelength of 488 nm and an emission wavelength of 525 nm.

## 2.15. scRNA-seq

Raw data for GSE178265 were obtained from the GEO database [14]. The R package "Seurat" was used to filter data. We filtered low-quality cells with the following criteria: genes were filtered that are only expressed in three cells or less, the cells were filtered by gene counts more than 5000 or less than 400, remove the cells with over 10% mitochondrial content. The "DoubletFinder" was used to remove the doublet cells. After the filtering, the functions "Seurat" was used for dimension-reduction and clustering. The logarithmic normalization method of the "Normalization" function was used to normalize and merge the expression of genes. Then, scaling analysis and PCA were performed. Using the top 10 principle components and Louvain algorithm, the cells were clustered into multiple clusters (The first 20 principle components were selected from the data of multi-sample integration). The t-distribution random neighborhood embedding (t-SNE) algorithm was applied to visualize cells in a two-dimensional space. Based on the Wilcox algorithm, the marker genes were identify. The cell type were annotated of each cluster according to the expression of canonical markers from

literatures. Finally, the "Seurat FeaturePlot" and "Vlnplot" function were used to display the location and expression pattern of feature gene in different cell types.

## 2.16. Statistical analysis

Data were shown as mean ± standard deviation (SEM). The Shapiro-Wilk test were used to assess the normality and the Brown-Forsythe test were used to assess the variance homogeneity. Paired sample t test and one-way ANOVA test were used to calculate the statistical significance (p-value) by the GraphPad Prism 8.0 and the $p < 0.05$ were considered to be statistically significant.

# 3. Results

## 3.1. Identification and clustering of ferroptosis-related genes

The study flowchart is shown in (**Fig 1**). Based on the four datasets, 97 RNAs related to substantia nigra were obtained from 54 PD patients and 43 controls. DEG analysis was conducted to identify the genes involved in the process of ferroptosis. We downloaded 564 ferroptosis genes from the FerrDb database and intersected with 453 DEGs identified between PD and control samples. After the multi-annotated genes were screened, 11 FRGs were identified, among which 8 genes were upregulated and 3 were downregulated (**Fig 2A and 2B**). A related heatmap of the collection of 11 genes is shown in **Fig 2C**. These labeled genes were verified by the boxplot. Six FRGs (*PML*, *SIAH2*, *KEAP1*, *SOX2*, *RELA*, and *SLC3A2*) were significantly upregulated and two FRGs (*SCP2*, *GRIA3*) were downregulated (**Fig 2D–2K**).

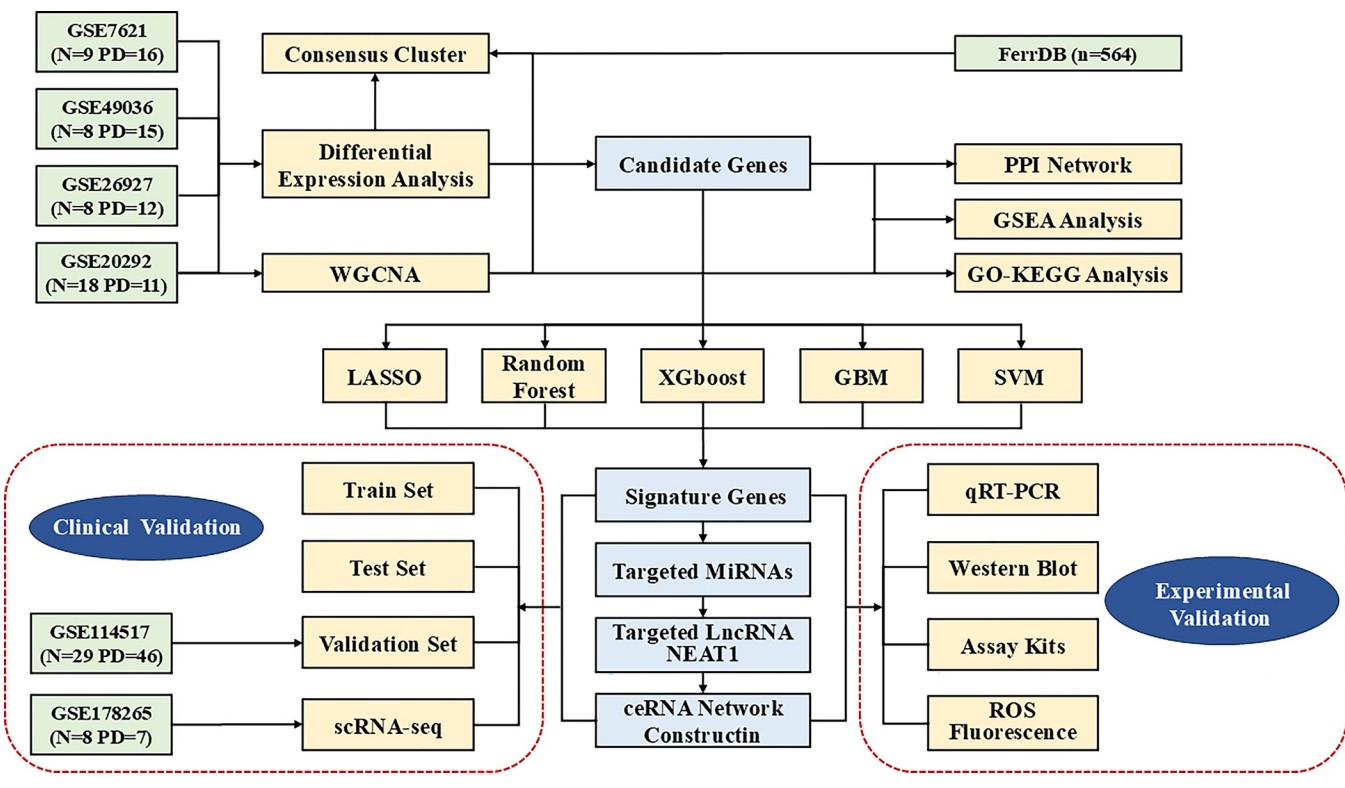

**Fig 1. Flowchart of this research.**

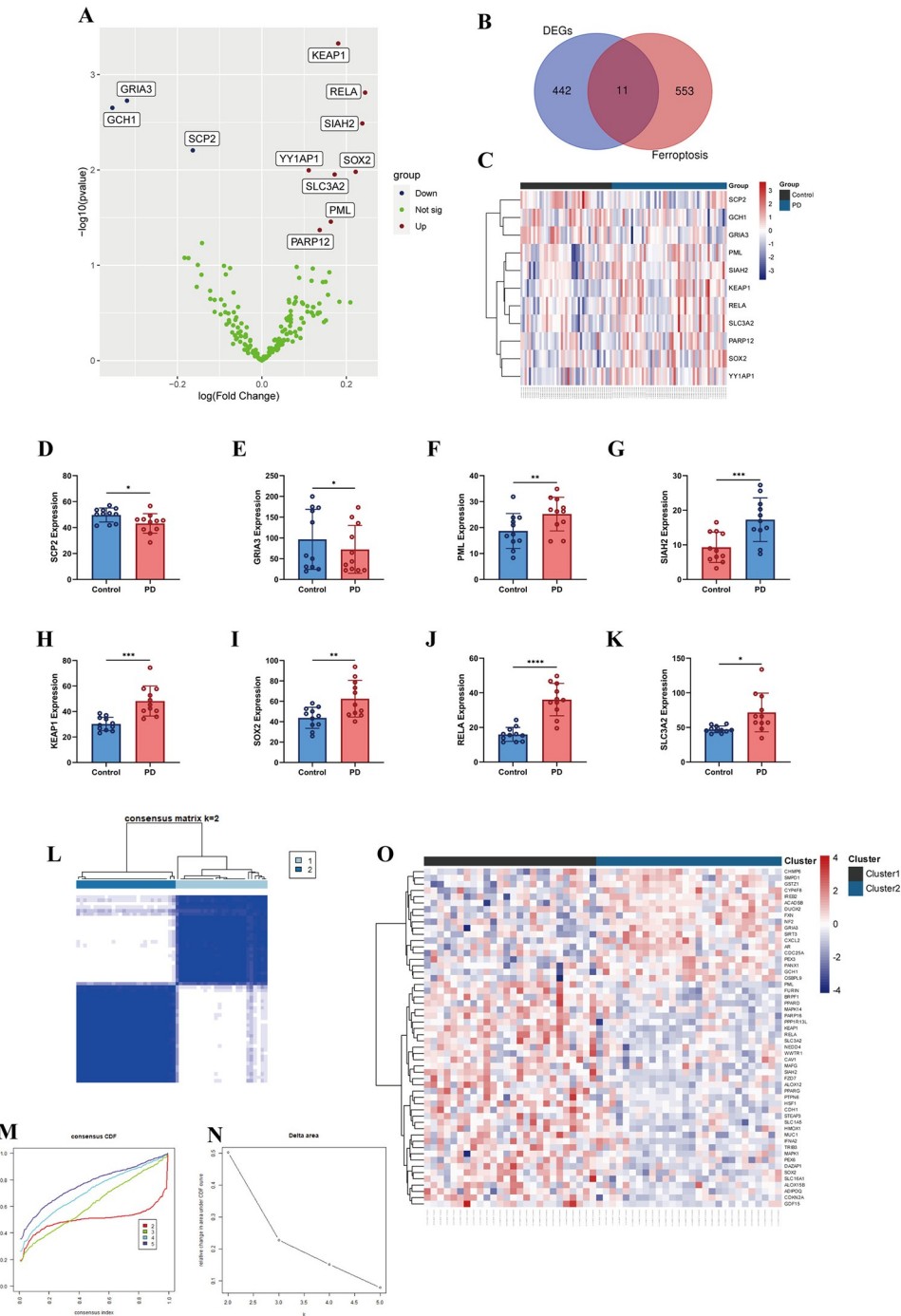

**Fig 2. Expression profile of DEGs and consensus clustering of FRGs.** (A) Volcano plot of FRGs between PD vs Control. (B) Venn diagram of FRGs between DEGs and FerrDB. (C) Heatmap of FRGs in PD. (D-K) Validation the expression of FRGs. (L) Consensus clustering at the index k = 2. (M) CDF of clustering (k = 2–5). (N) Delta area under the CDF curve. (O) Heatmap of FRGs in Clusters.

To further explore the unique expression characteristics of FRGs between individual PD patients, we used a consensus clustering algorithm based on the expression profiles of hub FRGs. k = 2 appeared to have the best stability and reliability for sorting the entire cohort into

two subtypes, including cluster 1 and cluster 2 (cluster 1 = 26, cluster 2 = 28) (**Fig 2L–2N**). The heatmap shows the expression profile of FRGs among the two clusters (**Fig 2O**).

### 3.2. Identification of DEGs based on FRGs

To explore the FRG function of each pattern, 453 DEGs were identified in 54 PD patients matched with 43 control samples using the R package "limma". As shown in **Fig 3A**, there were 226 upregulated genes and 227 downregulated DEGs. Moreover, based on the FRGs, 1663 DEGs were identified between cluster 1 and cluster 2 of PD samples, including 666 upregulated genes and 997 downregulated genes (**Fig 3B**). The hub DEGs with the intersection of the two clusters were identified, and the top 100 are depicted in the heatmap (**Fig 3C**).

### 3.3. Gene set enrichment analysis

GSEA was used to identify the two groups' biological functions and pathways. KEGG analysis showed that the MAPK signaling pathway, lysosome pathway, pathways in cancer, chemokine signaling pathway, endocytosis, and regulation of actin cytoskeleton were significantly enriched in the PD samples (**Fig 3D–3I**). Subsequently, the results indicated that DEGs between the two clusters were mainly enriched in the calcium signaling pathway, regulation of actin cytoskeleton, dilated cardiomyopathy, neuroactive ligand-receptor interaction, ubiquitin-mediated proteolysis, and purine metabolism (**Fig 3J–3O**). Further pathways such as melanoma, focal adhesion, prostate cancer, and some biological processes involved in neuron degeneration were also enriched.

### 3.4. Determination of hub modules in WGCNA

The application of WGCNA network analysis was established based on the GSE20292, containing the age-matched 11 PD and 11 normal samples selected to accurately identify DEGs. The independence degree was $\geq 0.85$, and the soft threshold power of 8 was selected to carry out scale-free networks (**Fig 4A and 4B**). The expression values of 5665 genes were utilized for cluster analysis and to detect the hub modules. We screened 8 key modules based on MEDiss Thres = 0.4 according to similar expression clinical traits (**Fig 4C**). Among the harvested modules, the green and the dark green modules were significantly associated with the characteristics of PD; thus, they were selected as the hub modules (**Fig 4D**). With the cutoff criteria $|GS| \geq 0.7$ and $|MM| \geq 0.8$, we obtained the 432 key genes shared with the green module (Cor = 0.93, p = 5e-10) (**Fig 4E**) and the 415 hub genes by the dark green module (Cor = -0.89, p = 3e-08) (**Fig 4F**). Accordingly, we focused on 847 hub genes associated with PD identified through the hub modules of WGCNA.

### 3.5. Functional and pathways enrichment analysis

To elucidate the latent biological functions and pathways associated with the risk of PD, we performed GO and KEGG enrichment analysis in the hub modules. GO terms of molecular function suggested that those genes are significantly involved in positive regulation of cytosolic calcium ion concentration, muscle contraction mononuclear cell proliferation, lymphocyte or leukocyte proliferation, and some immune-related biological processes (**Fig 4G**). Analysis of KEGG pathways indicated that the hub genes associated with PD are involved in neuroactive ligand-receptor or cytokine receptor interaction, hematopoietic cell lineage, complement, and coagulation cascades, as well as JAK-STAT signaling pathway (**Fig 4H**). These pathways suggest that neuroinflammation and cell signal transduction may participate in ferroptosis in PD patients.

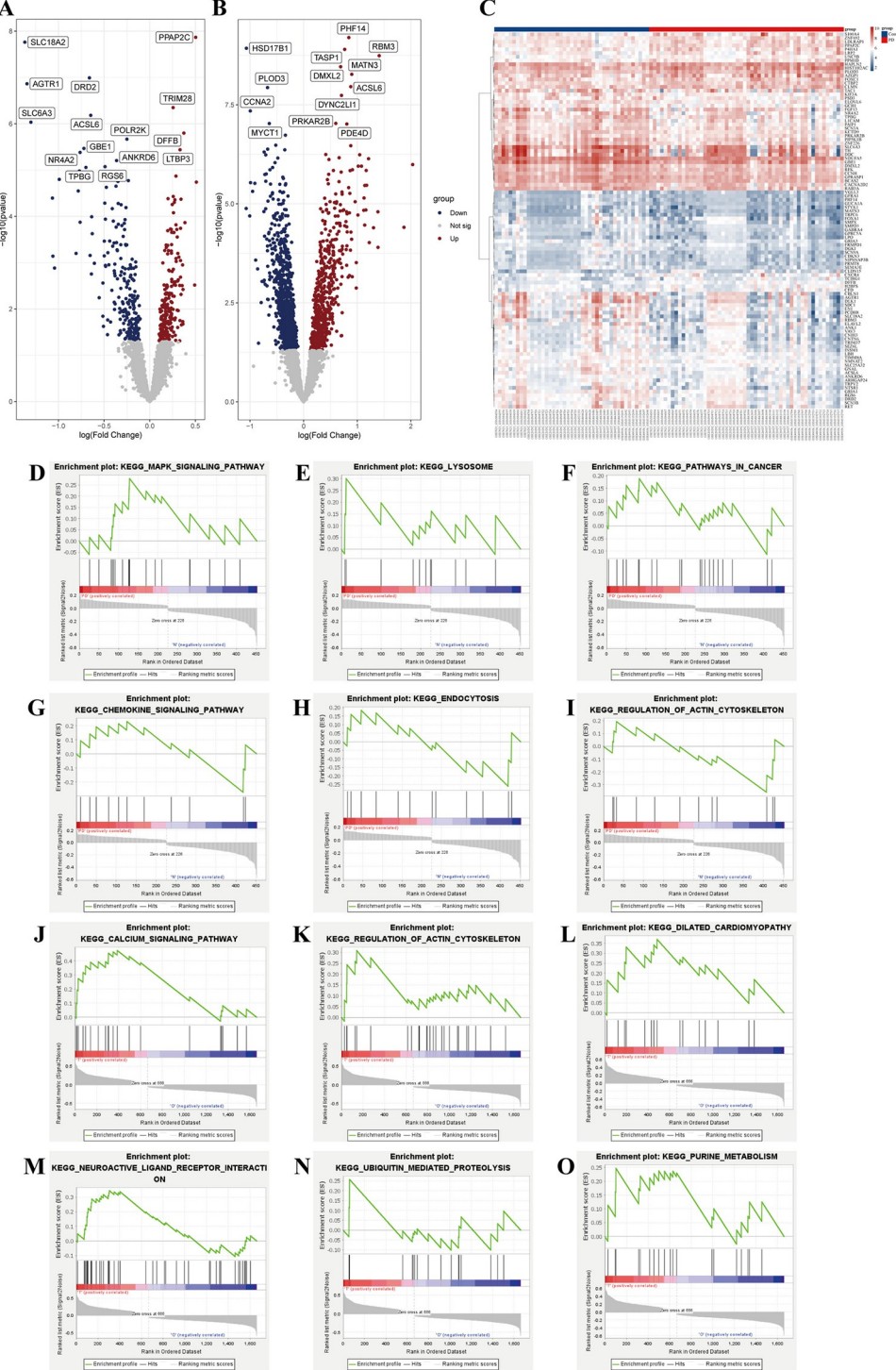

**Fig 3. Identification of DEGs and GSEA in two groups.** (A) Volcano plot of DEGs between PD vs Control. (B) Volcano plot of DEGs between cluster1 vs cluster2. (C) Heatmap of top 100 DEGs in two groups. (D-I) Biological functions and pathways of genes between PD vs Control. (J-O) Biological functions and pathways of genes between cluster1 vs cluster2.

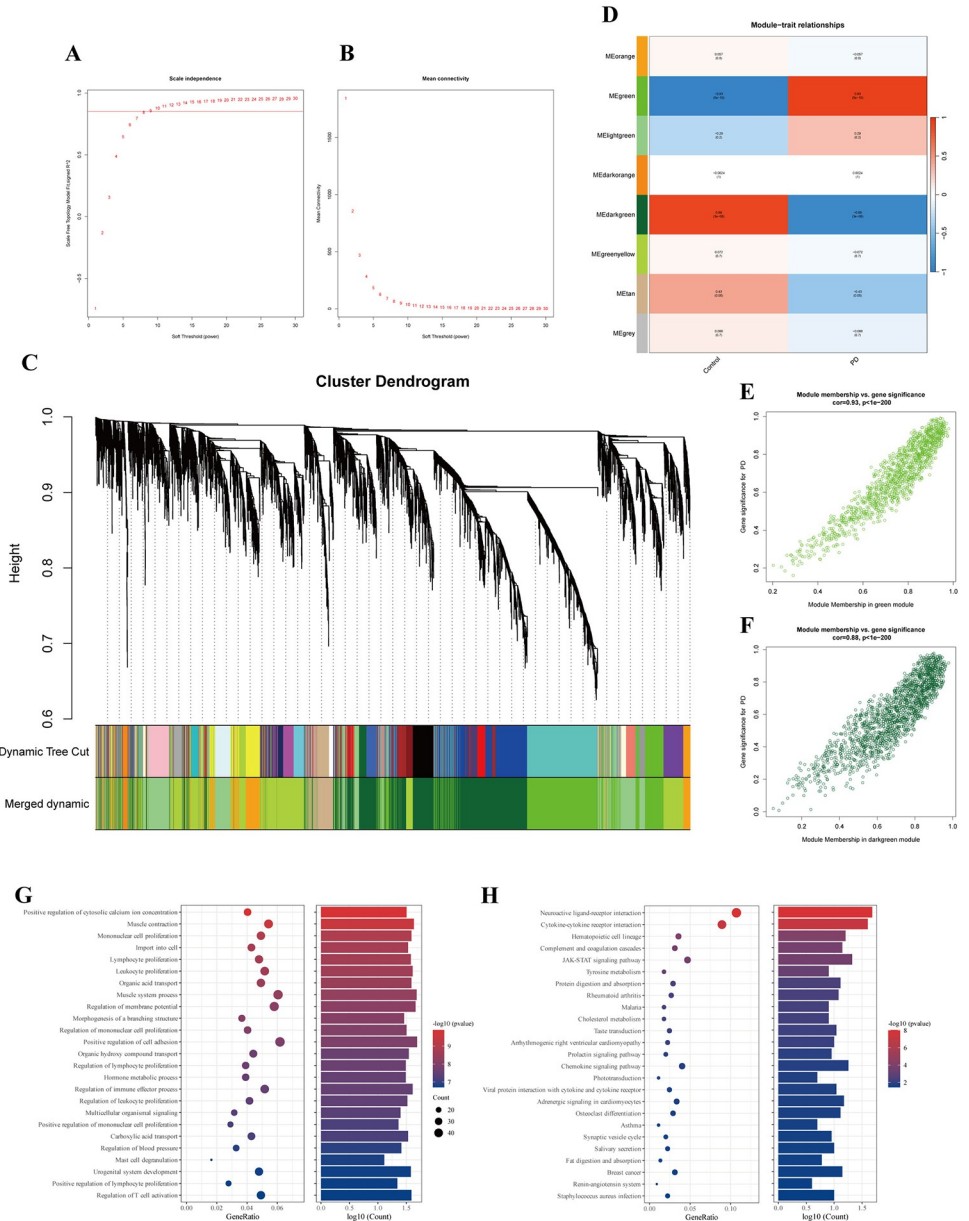

**Fig 4. WGCNA and enrichment analysis.** (A,B) Estimation of the independence degree and soft threshold power. (C) Cluster dendrogram of DEGs. (D) Correlation between modules and phenotypes of PD. (E,F) Scatterplot of GS vs MM in the green and darkgreen. (G) GO enrichment analysis of hub genes. (H) KEGG enrichment analysis of hub genes.

## 3.6. Multiple algorithms identify the hub genes-related PD classifier

A total of 67 hub genes were obtained from the DEGs of the three groups (PD vs. NC, cluster1 vs. cluster2, WGCNA) intersection (**Fig 5A**). The PPI network shows these hub genes' interactions (**Fig 5B**). To explore the signature genes associated with ferroptosis in the occurrence of PD, we found the diagnostic classifiers with five unique algorithms, including LASSO, random forest, XGBoost, GBM, and SVM. The feature choice strategy aimed to cut down the number of related genes (**S2 Table**). The LASSO regression was used to ascertain 12 key genes from the PD-related genes (**Fig 5C and 5D**). We selected the 27 hub genes from the top 30 candidate genes with the result of random forest analysis (**Fig 5E**). The SVM algorithm to filtrate

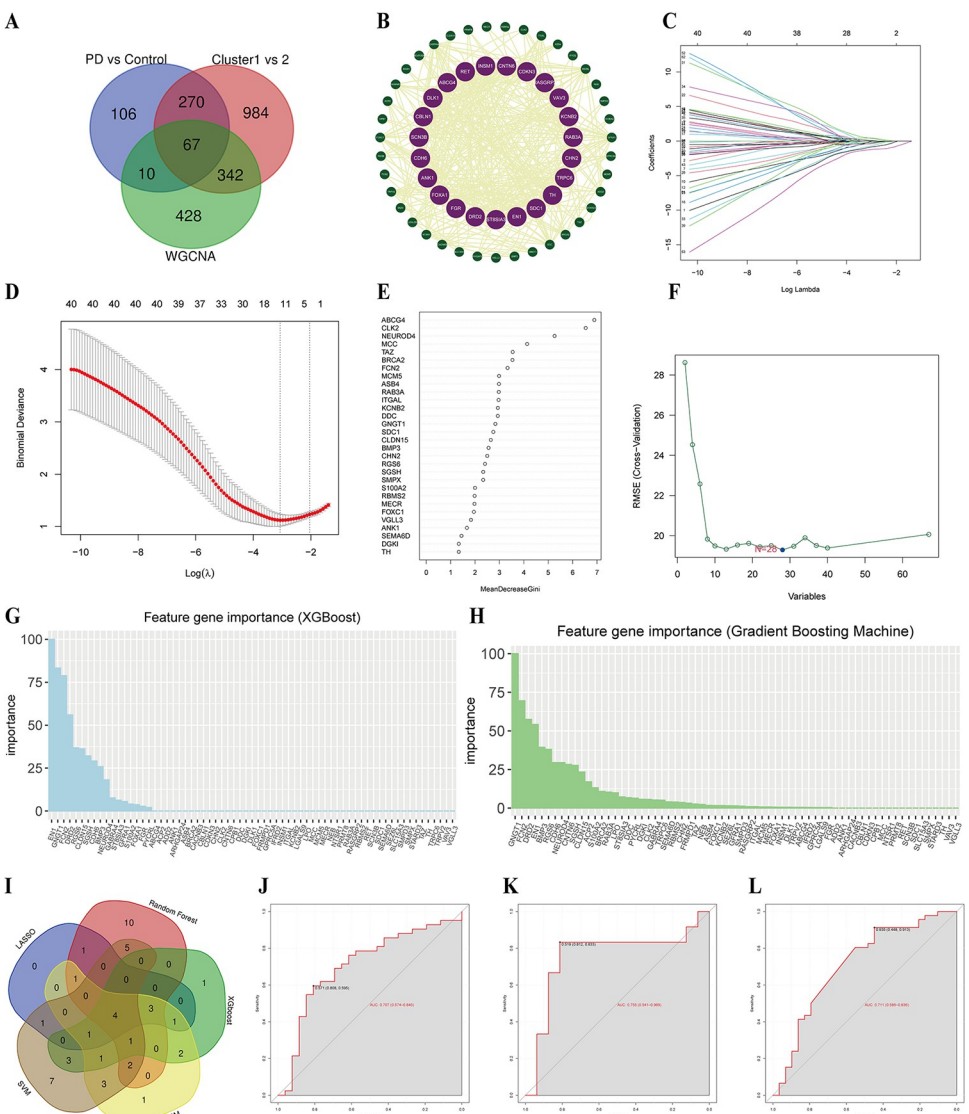

**Fig 5. Machine learning and construction of PD diagnostic model.** (A) Venn diagram of hub genes. (B) PPI network construction base on the hub genes. (C,D) LASSO coefficient profiles and cross validation for turning parameter(λ) of hub genes. (E) Hub genes selected from RF algorithm. (F) Screening conditions of SVM model. (G,H) Hub genes calculated by XGBoost and GBM algorithm and automatically rank. (I) Venn diagram of 4 features in five machine learning algorithms. (J-L) ROC curves shown the diagnostic value of 4 signature genes in training set, test set and external validation set.

irrelevant genes and 28 indicators were gained (**Fig 5F**). XGBoost model algorithm was then applied, revealing the top 17 ranked genes (Importance score > 2) as the main features (**Fig 5G**). Similarly, the GBM model chose the essential signatures, and the top 20 ranked genes (Importance score > 5) were selected (**Fig 5H**). Accordingly, 4 features (i.e., *S100A2*, *GNGT1*, *NEUROD4*, and *FCN2*) were obtained based on the intersection of five algorithms, as shown in the Venn diagram (**Fig 5I**).

## 3.7. Construction of PD diagnostic model

The predictive model was learned from the merged datasets, including 54 PD samples and 43 controls. To evaluate the overall prediction accuracy of the model, the training data was

randomly generated by selecting 70% of the dataset, and the remaining 30% was used as the test dataset. We also used NGS data GSE114517 as the validation dataset. The multivariable logistic regression analysis was used to assess hub features. The risk score of PD was calculated by the formula of Exp*GNGT1* × (-0.7124) + *ExpFCN2* × (-0.3334) + *ExpS100A2* × 0.6665 + Exp*NEUROD4* × (-0.4717). The ROC curve was drawn to assess the predictive accuracy of the PD diagnostic model by the AUC. The AUC of the training and test sets were 0.707 and 0.755, respectively (**Fig 5J and 5K**). Moreover, the AUC of the external validation set was 0.711 (**Fig 5L**). The ROC analysis results showed that the model had a higher diagnostic and predictive value in distinguishing PD patients from normal.

## 3.8. Validation of diagnostic signatures in PD ferroptosis model

The expression patterns of 4 features were verified in PD samples. It indicated that the *GNGT1*, *FCN2*, and *NEUROD4* were expressed at low levels, while *S100A2* was expressed at high levels in PD patients (**Fig 6A–6D**). Then, we constructed PD and ferroptosis models using SH-SY5Y cells, which were treated with erastin and 6-OHDA with or without Lip-1. The degeneration of dopamine was presented in the treatment of 6-OHDA. Compared with the control or erastin group, expression of TH was decreased in the 6-OHDA group. Nevertheless, Lip-1 inhibitor alleviated the 6-OHDA-induced loss of dopamine. To further affirm that ferroptosis exists in 6-OHDA treated SH-SY5Y cells, TFR, FTH1, ACSL4, and GPX4 expressions and iron, GSH, and MDA levels were detected. Compared with the control, 6-OHDA increased the expression of iron, TFR, ACSL4, and accumulation of MDA. The quantity of GSH-related pathways containing GPX4 and FTH1, was significantly decreased. Those biomarkers were all consistent with the erastin group and alleviated by the Lip-1 (**Fig 6E–6M**). Finally, the filtered 4 signature genes were verified using qRT-PCR. In the 6-OHDA groups, the expression levels of *S100A2* were visibly upregulated, and the *GNGT1*, *FCN2*, and *NEUROD4* were downregulated compared to control cells. Yet, Lip-1 reversed this process (**Fig 6N–6Q**), which suggested that the PD features model consisting of *GNGT1*, *FCN2*, *S100A2*, and *NEUROD4* is a useful auxiliary method for diagnosing PD, especially with neuron ferroptosis.

## 3.9. Construction of ceRNA network and validation of NEAT1 in PD ferroptosis model

Based on the 4 diagnostic signature genes, corresponding miRNAs and lncRNAs were predicted using the "NetworkAnalyst". Four miRNAs (*has-miR-335-5p*, *hsa-miR-7b-5p*, *has-miR-16b-5p*, and *has-miR-26b-5p*) with higher significance scores, cross-linking with at least 2 genes, were selected. Then, these 4 miRNAs acted as seed nodes. Next, 150 lncRNAs were obtained to construct the ceRNA network (**Fig 7A**). We extracted the lncRNAs with higher degrees and found that *NEAT1* interacted with both 4 miRNAs. The ceRNA network was then constructed, and the *NEAT1* was presumed to have important regulatory functions in the ferroptosis-related pathogenesis of PD (**Fig 7B**). In SH-SY5Y cells, 6-OHDA treatment increased the transcription of *NEAT1* and *S100A2*, the expression levels of *S100A2* could be well attenuated by using the si-NEAT1 (**Fig 7C and 7D**). Further, qRT-PCR results showed that the expression level of *miR-26b-5p*, targeting *S100A2*, was partially activated by si-NEAT1. However, there was no significant difference in *miR-7b-5p* levels with si-NEAT1 and si-NC (**Fig 7E and 7F**). Additionally, in the presence of 6-OHDA with si-NEAT1, *miR-7b-5p* inhibitor partially elevated the *S100A2* levels, indicating that *NEAT1* might competitively bind and regulate *S100A2* by suppressing *miR-26b-5p* (**Fig 7G**). We concluded that the *NEAT1* deficiency blocks the ferroptosis capacities of cells. Also, the knockdown of *NEAT1* rescued the down-regulated

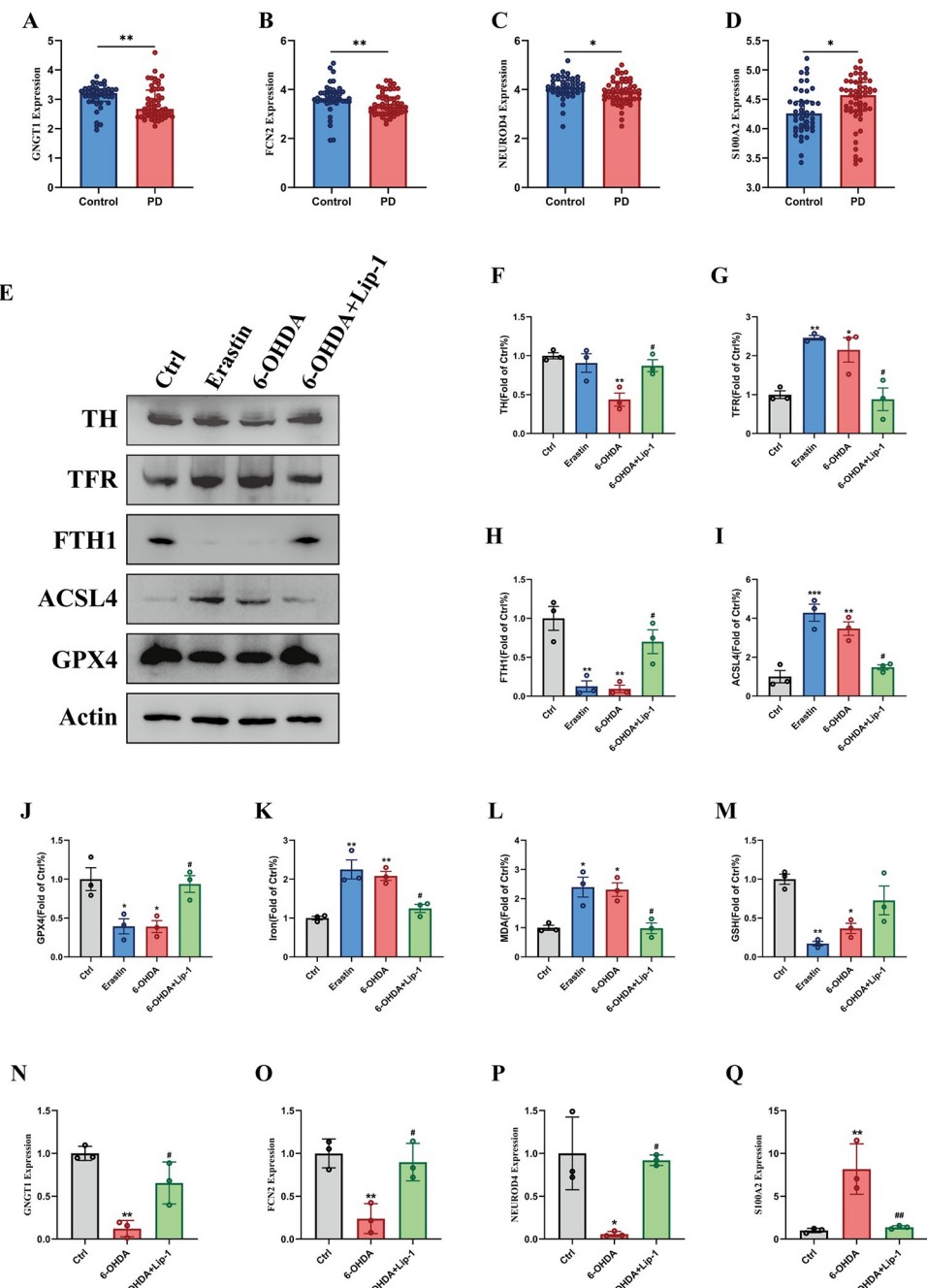

**Fig 6. Validation of the PD diagnostic model in vitro.** (A-D) The expression level of 4 features in PD. (E-J) Western blot analysis the expression of TH, TFR, FTH, ACSL4 and GPX4 in SH-SY5Y cells treated by the 6-OHDA. (K-M) The levels of iron, MDA and GSH in SH-SY5Y cells treated by the 6-OHDA. (N-Q) The mRNA relative expression levels of GNGT1, FCN2, S100A2 and NEUROD4 in SH-SY5Y cells. (N = 3, *p < 0.05 vs Ctrl group, **p < 0.01 vs Ctrl group, ***p < 0.001 vs Ctrl group. #p < 0.05 vs 6-OHDA group, ##p < 0.01 vs 6-OHDA group, ###p < 0.001 vs 6-OHDA group).

TH levels induced by the 6-OHDA. The fluorescence of DCFH-DA probe results showed that 6-OHDA exposure significantly raised the production of ROS and reduced cell viability (**Fig 7H and 7J**). In agreement with the above-mentioned, *NEAT1* knockdown made an inhibition effect on the ferroptosis damaging of 6-OHDA (**Fig 7I and 7K–7O**), which suggests that the

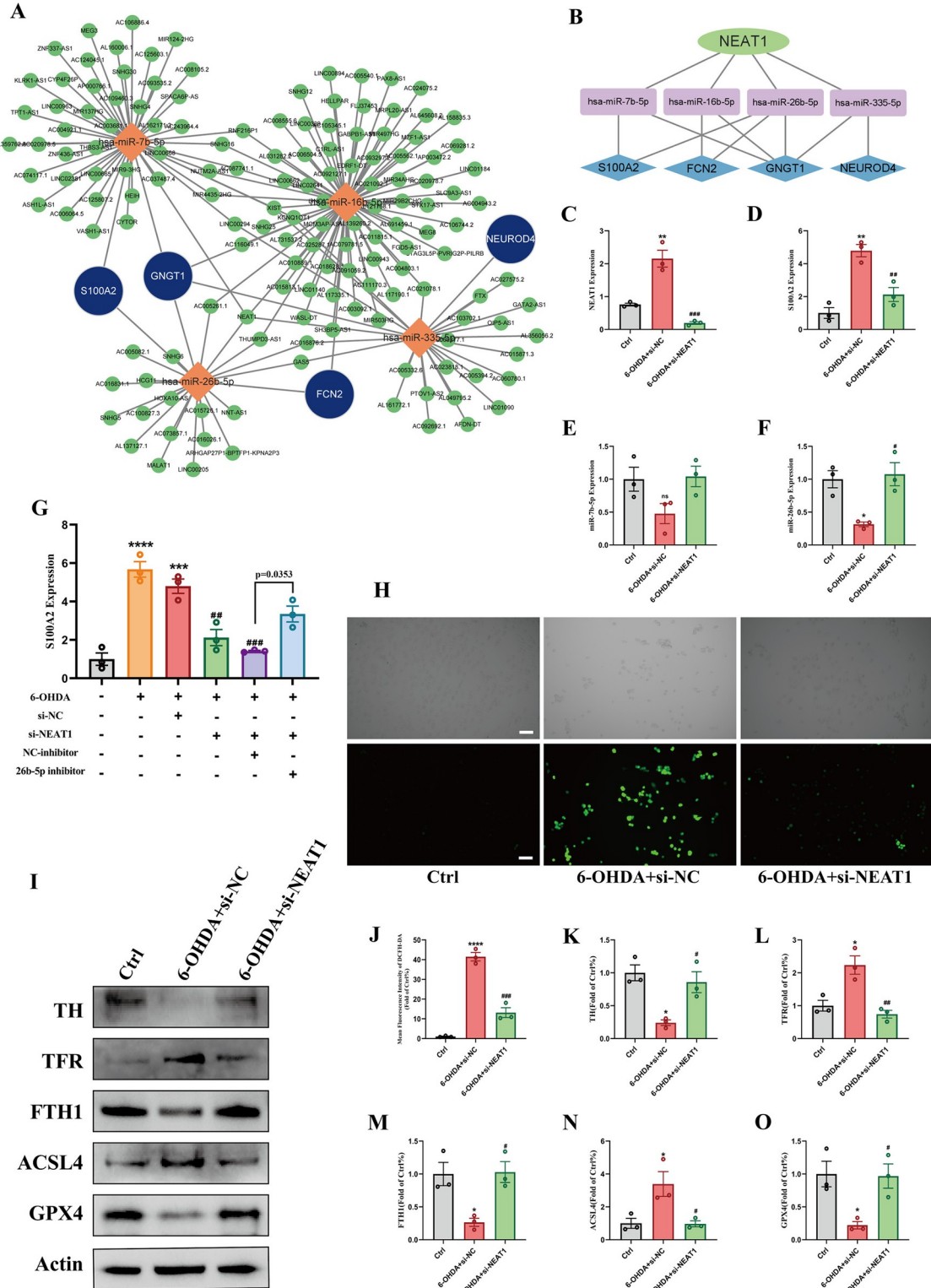

**Fig 7. Construction of ceRNA network and validation of NEAT1/miR-26b-5p/S100A2 axis in vitro.** (A) Prediction of miRNAs and lncRNAs based on the signature genes and construction of ceRNA network. (B) construction of primary ceRNA network based on the NEAT1. (C-F) The relative expression levels of NEAT1, S100A2, miR-26b-5p and miR-7b-5p in SH-SY5Y cells after NEAT1 knockdown. (G) The relative expression levels of S100A2 in SH-SY5Y cells after miR-26b-5p inhibition. (H, J) The levels of ROS were quantified by measuring the fluorescence of DCFH-DA in SH-SY5Y cells after NEAT1 knockdown, scale bar = 100μm. (I,

K-O) Western blot analysis the expression of TH, TFR, FTH, ACSL4 and GPX4 in SH-SY5Y cells after NEAT1 knockdown. (N = 3, *p < 0.05 vs Ctrl group, **p < 0.01 vs Ctrl group, ***p < 0.001 vs Ctrl group. #p < 0.05 vs 6-OHDA +si-NC group, ##p < 0.01 vs 6-OHDA+si-NC group, ###p < 0.001 vs 6-OHDA+si-NC group).

potential correlation between *NEAT1* and ferroptosis in PD and NEAT1/miR-7b-5p/S100A2 axis might participate in PD pathogenesis.

### 3.10. scRNA-seq analysis for the location and expression pattern of NEAT1

To further explore the cell diversity of NEAT1 in PD, the GSE178265, including seven PD samples and eight normal samples, was used to perform scRNA-seq analysis. After the preliminary quality control and standardization of gene expression, 37906 cells were derived from PD and normal samples (**Fig 8A**). The cells were clustered into 22 clusters (**Fig 8B**). Using the marker genes (**S2 Fig**), we classified 22 cell clusters into nine cell populations (**Fig 8C**). It showed that the nine main isolated cell groups comprising the microglial cells, astrocytes, OPCs, oligodendrocytes, DA neurons, NonDA neurons, endothelial cells, pericytes and fibroblasts (**Fig 8D**). Subsequently, the expression and localization in SN tissue between PD and normal samples were determined (**Fig 8E**). It was found that the NEAT1 were obviously expressed in all nine kinds of cells, and the expression level were both elevated (**Fig 8F**). Although the PD patients have significant loss in cell numbers and ratios of DA neurons, the expression level of NEAT1 in PD samples were still higher than control groups (**Fig 8G**). It was consistent with our result and verified that NEAT1 play an important role in the prognosis of PD.

## 4. Discussion

Evidence suggests that iron metabolism is closely related to the pathogenesis of PD and that multiple iron-regulatory proteins have potential diagnostic value in PD [15,16]. Neuroimaging and post-mortem examination showed that iron deposition in SN can promote α-synuclein, causing lipid peroxidation or producing ROS [17]. Moreover, few studies have shown that ferroptosis inhibitors could prevent ferroptosis and limit neurodegeneration in PD [18]. In this study, we developed a statistical diagnosis model of PD that accounts for the ferroptosis effects.

We comprehensively analyzed the expression profiles of pivotal FRGs associated with the pathogenesis of PD samples. Based on the five GEO datasets and the online FerrDB database, we screened out 11 differentially expressed FRGs containing 8 overexpression genes (*PARP12*, *PML*, *SLC3A2*, *YY1AP1*, *SOX2*, *SIAH2*, *RELA*, and *KEAP1*) and 3 downregulated genes (*SCP2*, *GCH1*, and *GRIA3*). Subsequently, 8 genes (i.e., *PML*, *SLC3A2*, *SOX2*, *SIAH2*, *RELA*, *KEAP1*, *SCP2*, and *GRIA3*) were verified by utilizing the NGS datasets. *PML* is a tumor suppressor that regulates mitochondrial ferroptosis in cancer, most probably through a stress-mediated PML-PGC-1α-dependent mechanism, enhancing ferroptosis sensitivity [19]. *SLC3A2* is the second most important part containing the xCT after *SLC7A11*, inhibiting GSH synthesis. In addition to the induction of ferroptosis, *SLC3A2* is also involved in the mechanism of levodopa absorption [20]. *SOX2* is well known for its roles in the differentiation and development of induced pluripotent stem cells; it can also block myelination in the Schwann cells with increased inflammation [21]. *SIAH2*, as an E3 ubiquitin-ligases, has a key role in monoubiquitylates α-synuclein or immunoreactivity in Lewy bodies [22]. *RELA* (p65) and *KEAP1* are the hub transcription factors of the NF-KB/NRF2 signaling pathway that regulate antioxidant elements in PD [23]. The gene *GRIA3* encodes the GluA3 subunit of the AMPA receptor, which

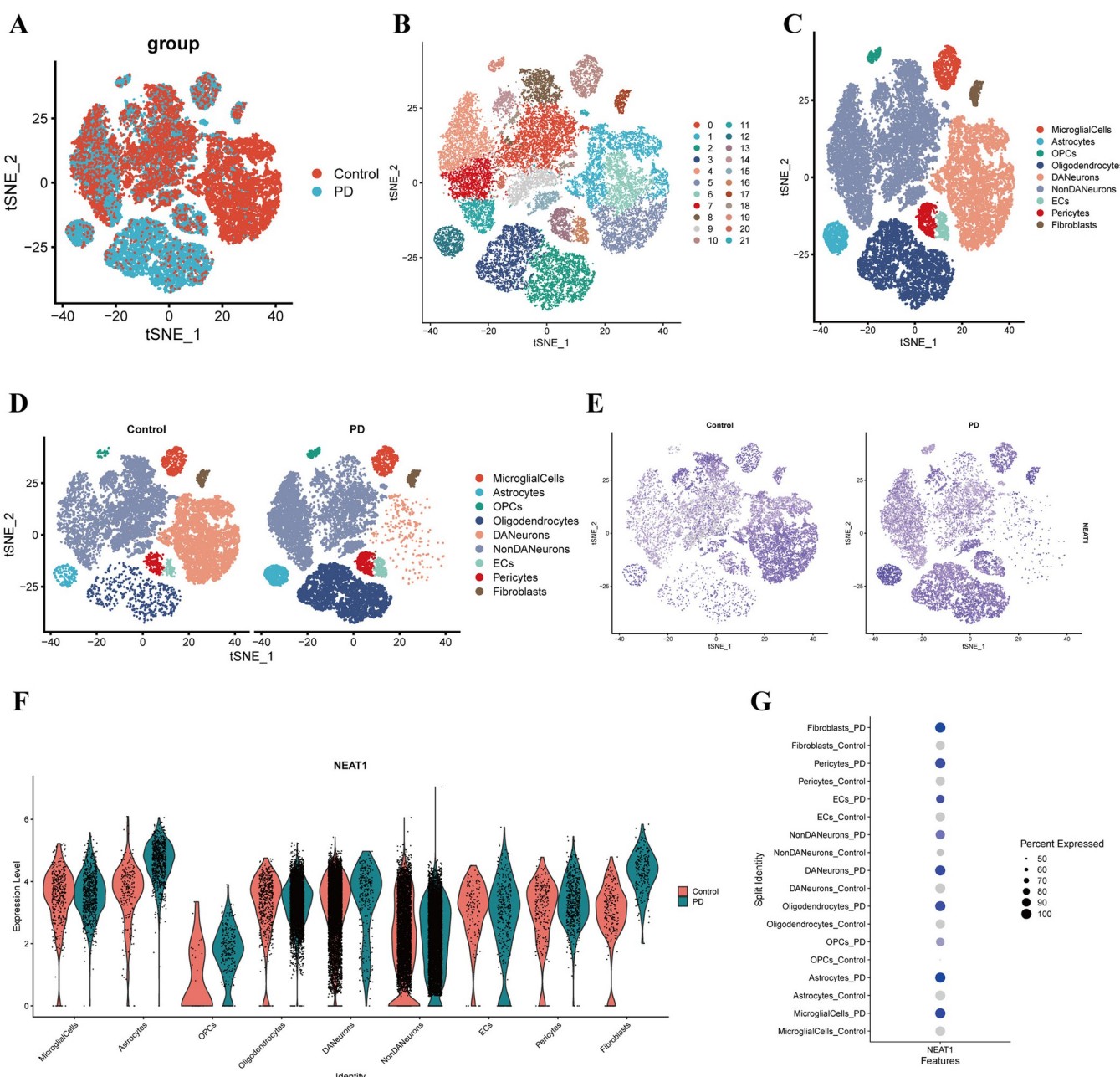

**Fig 8. scRNA-seq analysis of NEAT1.** (A) The cellular landscape of PD and normal samples. (B) TSNE map of 22 clusters. (C,D) TSNE map of nine cell populations classified by genetic marker in PD and normal samples. (E) Expression of NEAT1 at the cellular level of the control and PD group. (F) The expression level of NEAT1 in nine cell populations. (G) Dot plot of NEAT1 levels in cell populations between PD and control.

is widely associated with neurodevelopmental disorders and is a synaptic marker of neurodegenerative diseases [24]. All these FRGs are associated with PD-related ferroptosis.

We utilized the hub FRGs to cluster consensus and divided the PD samples into two groups. DEGs related to ferroptosis in PD are mainly involved in the MAPK signaling pathway, ubiquitin-mediated lysosome pathway, regulation of actin cytoskeleton, and neuroactive ligand-receptor interaction. According to WGCNA analysis, two hub modules were identified as significantly enriched in the calcium signaling pathway, leukocyte proliferation, migration,

immunity, and neurotransmitter transmission or metabolism. Most of those biological processes indicate that the ferroptosis in PD patients may be associated with neuroinflammation. Previous studies have reported the interaction between α-synuclein and microtubule protein, demonstrating the function of membrane structure in PD pathogenesis [25]. Moreover, the pathways mentioned above also participate in autophagy, another special programmed cell death that may overlap with ferroptosis in PD [26]. Our results reflect the basic molecular function of biological processes and the continuity of potential co-expression information.

In this study, 67 hub genes were selected from the intersection with three groups (PD vs. NC, cluster 1 vs. cluster 2, WGCNA). Subsequently, five machine learning algorithms were utilized to filtrate 4 potentially pivotal FRGs features (*S100A2*, *GNGT1*, *NEUROD4*, and *FCN2*) related to iron metabolism and ferroptosis in neurodegeneration. Then, a novel diagnostic 4-features model was constructed. This model may screen PD patients and be convenient for clinical work as it only contains 4 features.

*S100A2* is a gene coding for an important member of S100 protein that is implicated in varied functions and complex networks. As the regulatory of calcium-binding proteins, a mutation in *S100A2* affects cellular physiological functions, such as the downregulation of enzyme activities, calcium dyshomeostasis, and protein phosphorylation [27]. It has been reported that *S100A2* is involved in the pathogenesis of many cancers. However, the role of *S100A2* in tumors seems to be dual [28]. *S100A2* is also a hallmark of aging that mediates signal transduction in series of neurodegeneration. Up regulated the levels of *S100A2* altered the metal-buffering activity, such as cadmium, zinc, or iron, may reveal the underlying mechanism [29]. Previous studies proved that the *S100A2* could activate the PI3K/AKT signaling pathway, upregulate the GLUT1 expression, and promote glycolytic reprogramming. *S100A2* is also associated with the immune microenvironment and neuroinflammation by enhancing the IL-17 and TNF signaling pathways [30].

G protein subunit gamma transducin 1 is encoded by the *GNGT1*, which is highly expressed in the eye and is strongly associated with retinal defects. However, *GNGT1* transcript is also expressed in the lung, heart, alimentary canal, and skeletal muscle, where the signaling mechanism remains unclear [31]. *GNGT1* regulates cell proliferation, migration, adhesion, and differentiation in those tissues and induces apoptosis [32]. It remains unknown whether *GNGT1* is involved in the ferroptosis of neurodegenerative disease. Some studies found that *GNGT1* may be used as a biomarker of medulloblastoma, and its mutation may be related to neuroautoimmune diseases, such as multiple sclerosis [33,34], providing a new angle to evaluate the inherited causes of neurodegeneration.

*NEUROD4* is widely expressed in the nervous system. It encodes the xenopus protein, also known as Xath3 (Math3/NeuroM). The *NEUROD4* has an intermediate regulatory role at neural plate stages in the primary embryo and limits the ability to drive developing neurogenesis. Mechanistically, *NEUROD4*, as the Basic Helix Loop Helix (BHLH) proneural transcription factor, drives the transition between proliferation and differentiation with the regulation of autophosphorylation. The phospho-mutant *NEUROD4* can stabilize the proneural protein involved in the cell cycle [35]. Importantly, *NEUROD4* may promote neuroinflammation and the progression of oxidative stress in the central nervous system [36]. For another, the gene is essential to the progression of ASCL1-mediated astrocytes efficiently transform into neurons [37]. It may offer novel views into comprehending the molecular mechanisms underlying the differentiation of stem cells and nerve regeneration.

*FCN2* encodes the human ficolins that bind to specific pathogen-associated molecular patterns and show great potential in innate immunity to infectious diseases. In normal human plasma, *FCN2* acts as a lectin-complement pathway activator to recognize pathogens [38]. The role of *FCN2* gene polymorphisms and the level of *FCN2* in serum appear to be associated with

various bacterial or virus diseases, including mycobacterium tuberculosis, hepatitis virus, and dengue fever [39]. Moreover, *FCN2* also contributes to rheumatic and premature delivery as well as different kinds of cancers [40,41]. Overexpression of *FCN2* inhibits hepatocellular carcinoma through the TGF-β signaling pathway [42]. Few studies have reported the function of *FCN2* in neurons. Our data suggest that *FCN2* is a valuable marker for PD-related ferroptosis.

Herein, we developed a diagnostic model to distinguish PD based on the four ferroptosis-related features. In the previous study, several biomarkers have been explored to participate in the PD diagnosis, such as the α-synuclein in blood and CSF [43]. Moreover, combined with advanced medical imaging methods, they may be of particular interest in prodromal PD diagnosis and be able to predict the occurrence in PD patients [44]. However, most of the research is still focused on the clinic area, incremental evidence has highlighted the significance of genetic and biological factors in PD diagnosis recently. Part of PD immune infiltration patterns has been constructed to identify immune-related diagnostic biomarkers [45]. Based on the relationship between iron and PD, we selected the model genes related to ferroptosis, aiming to fill the gap in genetic biomarkers in PD. Our results indicated that the new model had certain diagnostic and prominent individual predictive effects.

To further confirm our diagnostic model, we conducted an in vitro PD cell model using 6-OHDA, a neurotoxin extensively used to induce the death of dopaminergic neurons in PD. The 6-OHDA induced ferroptosis in the human dopaminergic SH-SY5Y cells model, which is consistent with erastin groups and other studies [46]. Data showed that the biomarkers of ferroptosis, TFR, ACSL4, iron, and MDA levels were significantly upregulated in the 6-OHDA group, whereas FTH1, GPX4, and GSH were downregulated compared to control. The 6-OHDA also stimulates the loss of TH and dopamine, it also reduces *NEUROD4*, *GNGT1*, and *FCN2* mRNA while increasing *S100A2* mRNA. Yet, after treating cells with Lip-1, a particular ferroptosis inhibitor, cell damage was alieved and the change of hub genes caused by 6-OHDA was suppressed. Lip-1 can trap radicals and slow the accumulation of lipid hydroperoxides in PD samples [47]. Taken together, we proposed that the 4 signature genes (*NEUROD4*, *GNGT1*, *FCN2*, and *S100A2*) may be the essential contributors to the progression of PD-related ferroptosis, thus suggesting that targeting the diagnostic model may be an exploitation strategy for PD treatment.

Then, the original ceRNA network was constructed, elucidating the hypothesis that *NEAT1* targeting miRNAs might negatively correlate with diagnostic signature genes. *NEAT1* is considered an important lncRNA associated with the proliferation or migration of tumor cells. For example, a recent study suggested that high expression of *NEAT1* may induce ferroptosis by regulating *miR-362-3p* in the therapeutic strategy for hepatocellular carcinoma [48] and it is also considered to be a novel diagnostic biomarkers for Alzheimer's disease [49]. Our studies have shown that *NEAT1* may directly or indirectly target the 4 signature genes, especially *S100A2*. The expression trend of *NEAT1* and *S100A2* is consistent with the 6-OHDA induction. Knockdown the *NEAT1* also inhibited the *S100A2*. However, the expression of mRNA and lncRNA were negatively correlated with the *miR-26b-5p*, which acts as the connection point between *S100A2* and *NEAT1*. These data indicate that the *miR-26b-5p* may have essential regulatory roles in this axis. The *miR-26b-5p* has been reported to be involved in mitochondrial dynamics and predicted to be a specific biomarker of Alzheimer's disease [50,51], but the effect on *NEAT1* or *S100A2* has not yet been studied. We concluded that *miR-26b-5p* is a direct target of *NEAT1*, and the *S100A2* is its specific downstream signaling protein. Functionally, transfection of *miR-26b-5p* inhibitor partially reversed the downregulation of *S100A2*, which was caused by the si-NEAT1. Like the effect of Lip-1, si-NEAT1 could counteract the change of iron-related transporters, abrogate the ROS, and elevate the expression of TH, consequently delaying the occurrence of ferroptosis. According to the scRNA-seq results, We also found

that the NEAT1 were highly expressed in PD, while being more pronounced in astrocytes, oligodendrocytes and fibroblasts. It revealed that NEAT1 upregulation might promote to MPP+-induced neuron inflammation via NEAT1-miR-1277-5p-ARHGAP26 pathway [52]. Notably, even though dopaminergic neurons are severely depleted in PD group, we still observed that the expression of NEAT1 remained significantly higher than the control group. This is consistent with many previous findings regarding the role of NEAT1 in inducing autophagy, apoptosis, cytotoxicity, oxidative stress in PD mouse or cell models [53]. Even though the latent biologic mechanism is still undefined, we have reason to believe that *NEAT1* affecting the development of ferroptosis and the NEAT1/miR-26b-5p/S100A2 axis may be associated with this independent model of death in PD patients.

This study has several limitations. On one hand, the datasets used in our research are based on the public databases with a limited sample size. Different databases or threshold criteria may lead to different results. On the other hand, we only validated our findings through in vitro experiments. Exploring deeper molecular biological mechanisms through in vivo or stem cell experiments is necessary in the future. Furthermore, the feature genes were derived from post-mortem brain samples. More research should validate them in easily accessible samples such as blood, urine, and cerebrospinal fluid to facilitate clinical application. In conclusion, this study profound significance for exploring the mechanism of ferroptosis regulated by NEAT1/miR-26b-5p/S100A2 axis in PD. Further experiments are needed to confirm our findings, and more comprehensive genomic and normative clinical information should be performed. In conclusion, our findings might provide new insights into improving the understanding of the mechanism for ferroptosis in the prevention of PD and the potential therapeutic targets for timely symptomatic treatment.

## Supporting information

**S1 Table. Primers used for quantitative qRT-PCR.**
(DOCX)

**S2 Table. Results of machine learning.**
(DOCX)

**S1 Fig. Batch correction and quality control.** (A) Background corrected and normalized the batch effect of datasets. (B) PCA analysis of the four datasets after the normalized.
(TIF)

**S2 Fig. Dot plot of the marker genes in cluster analysis.**
(TIF)

**S1 Raw images.**
(PDF)

## Author Contributions

**Conceptualization:** Jifeng Guo.

**Data curation:** Taole Li.

**Formal analysis:** Taole Li.

**Funding acquisition:** Jifeng Guo.

**Methodology:** Taole Li.

**Project administration:** Jifeng Guo.

**Resources:** Jifeng Guo.

**Software:** Taole Li.

**Supervision:** Jifeng Guo.

**Validation:** Taole Li.

**Visualization:** Taole Li.

**Writing – original draft:** Taole Li.

**Writing – review & editing:** Taole Li, Jifeng Guo.

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
