## [Decision Letter · Decision Letter 0]

23 Sep 2024

PONE-D-24-16545

Identifying the NEAT1/ miR-7b-5p/S100A2 Axis as a Regulator in Parkinson’s Disease based on the Ferroptosis-Related Genes

PLOS ONE

Dear Dr. Guo,

Thank you for submitting your manuscript to PLOS ONE. After careful consideration, we feel that it has merit but does not fully meet PLOS ONE’s publication criteria as it currently stands. Therefore, we invite you to submit a revised version of the manuscript that addresses the points raised during the review process.

Reviewers have pointed out the lack of details and rationale for some methods. For example, the definition of log2FC Threshold in 0.1. Please, add all details (Sequences, parameters, catalog IDs, suppliers, etc) required to do this research reliable and reproducible. Parkinson disease is a in-development research field. Therefore, It Is relevant to validate your findings using external data. In this context, using single-cell analyses should help to understand the intrasample heterogeneity and cell-to-cell communication systems. However, the proper reasons and suitability of chosen single-cell dataset must be clearly described.

We look forward to receiving your revised manuscript.

Kind regards,

Alexis G. Murillo Carrasco

Academic Editor

PLOS ONE

Journal requirements: 1. When submitting your revision, we need you to address these additional requirements. Please ensure that your manuscript meets PLOS ONE's style requirements, including those for file naming. The PLOS ONE style templates can be found at https://journals.plos.org/plosone/s/file?id=wjVg/PLOSOne_formatting_sample_main_body.pdf and https://journals.plos.org/plosone/s/file?id=ba62/PLOSOne_formatting_sample_title_authors_affiliations.pdf. 2. PLOS ONE now requires that authors provide the original uncropped and unadjusted images underlying all blot or gel results reported in a submission’s figures or Supporting Information files. This policy and the journal’s other requirements for blot/gel reporting and figure preparation are described in detail at https://journals.plos.org/plosone/s/figures#loc-blot-and-gel-reporting-requirements and https://journals.plos.org/plosone/s/figures#loc-preparing-figures-from-image-files. When you submit your revised manuscript, please ensure that your figures adhere fully to these guidelines and provide the original underlying images for all blot or gel data reported in your submission. See the following link for instructions on providing the original image data: https://journals.plos.org/plosone/s/figures#loc-original-images-for-blots-and-gels.   In your cover letter, please note whether your blot/gel image data are in Supporting Information or posted at a public data repository, provide the repository URL if relevant, and provide specific details as to which raw blot/gel images, if any, are not available. Email us at plosone@plos.org if you have any questions. 3. Please note that PLOS ONE has specific guidelines on code sharing for submissions in which author-generated code underpins the findings in the manuscript. In these cases, we expect all author-generated code to be made available without restrictions upon publication of the work. Please review our guidelines at https://journals.plos.org/plosone/s/materials-and-software-sharing#loc-sharing-code and ensure that your code is shared in a way that follows best practice and facilitates reproducibility and reuse. 4. We note that the grant information you provided in the ‘Funding Information’ and ‘Financial Disclosure’ sections do not match.  When you resubmit, please ensure that you provide the correct grant numbers for the awards you received for your study in the ‘Funding Information’ section. 5. We are unable to open your Supporting Information file [raw data.zip]. Please kindly revise as necessary and re-upload. 6. Please include captions for your Supporting Information files at the end of your manuscript, and update any in-text citations to match accordingly. Please see our Supporting Information guidelines for more information: http://journals.plos.org/plosone/s/supporting-information. 

Reviewers' comments:

Reviewer's Responses to Questions

Comments to the Author

1. Is the manuscript technically sound, and do the data support the conclusions?

Reviewer #1: Partly

Reviewer #2: Yes

Reviewer #3: Partly

2. Has the statistical analysis been performed appropriately and rigorously?

Reviewer #1: I Don't Know

Reviewer #2: Yes

Reviewer #3: Yes

3. Have the authors made all data underlying the findings in their manuscript fully available?

Reviewer #1: Yes

Reviewer #2: Yes

Reviewer #3: No

4. Is the manuscript presented in an intelligible fashion and written in standard English?

Reviewer #1: Yes

Reviewer #2: Yes

Reviewer #3: Yes

5. Review Comments to the Author

Reviewer #1: The study has significant issues with scientific reasoning, research methods, and data analysis. The primary concern is using the sva package to remove batch effects, which is not recommended. Instead, batch should be included as a covariate in statistical models. The manuscript at https://doi.org/10.1093/biostatistics/kxv027 is highly recommended for further guidance. Programs like ComBat aim to modify the data to eliminate batch effects, but it is better to model the batch effect in the design formula without directly modifying the data.

Furthermore, the methodology section lacks clarity and detail in certain areas. For instance, the specific methods used to correct the background and normalize the data are unclear. It is also uncertain whether any filters were applied to exclude probes with no expression or if quality control measures such as principal component analysis were conducted. Additionally, the authors do not appear to have addressed potential confounding factors, and it seems that multiple testing corrections were not utilized. The rationale for selecting |log2 fold change | > 0.1 as the threshold for differential expression analysis is unclear. The authors should also clarify the reasoning behind the selection of datasets, as the WGCNA analysis was carried out using only one dataset, while the DEGs analysis involved different datasets.

The authors are strongly encouraged to review the "Minimum Information for Publication of Quantitative Real-Time PCR Experiments" (MIQE) guidelines. The MIQE publication provides detailed information on the minimal and desirable reporting requirements for qPCR data publication.

Reviewer #2: The authors investigated the pathogenesis of PD by identifying ferroptosis-related genes (FRGs). These FRGs were used to construct a competing endogenous RNA (ceRNA) network through bioinformatics analysis, complemented by experimental validation. The authors propose that lncRNA NEAT1 plays a crucial role as a biological regulator of ferroptosis in the onset and progression of PD. Additionally, they explored the regulatory axis NEAT1/miR-26b-5p/S100A2 in ferroptosis PD models through experimental work.

The manuscript is well-written and the scope is clearly defined. The data from the bioinformatics analysis are timely and significant for identifying novel markers to study the etiology of PD and to eventually develop targeted therapeutic strategies to slow its progression.

The work is interesting, and the authors’ effort in interrogating and integrating multiple datasets, along with including experimental validation, makes it suitable for publication in PLOS ONE.

Minor comments for manuscript preparation before publication are listed below:

- The manuscript would benefit from including single-cell expression analysis of NEAT1, NEUROD4, GNGT1, FCN2, and S100A2 in single-cell human post-mortem tissue (Kamath et al., Nature Neuroscience, 2022) and in single-cell midbrain organoids (Fiorenzano et al., Nature Communications, 2021). This addition would not only strengthen the authors’ findings but also highlight the potential use of brain organoids in future studies to explore the identified regulatory axis in more physiologically relevant conditions.

- The Materials and Methods section, particularly regarding the experimental use of 6-OHDA, requires further detail and clarification to improve reproducibility and transparency.

- The use of SH-SY5Y cells may not fully capture the complexity of the identified axis. It would be beneficial for the authors to show tyrosine hydroxylase (TH) expression and potential fragmentation at the immunofluorescence level before and after 6-OHDA treatment. This would provide a more comprehensive evaluation of the cellular model and the relevance of the regulatory pathway under study.

Reviewer #3: Thank you for submitting your manuscript. After careful review, I have the following comments and suggestions:

[Technical Soundness and Data Support]

1. The machine learning model validation process needs more detail. While you mention using training, test, and external validation sets, the manuscript lacks information on cross-validation procedures, which are crucial for ensuring model robustness.

2. The biological significance of the identified NEAT1/ miR-7b-5p/S100A2 axis in PD were verified in SH-SY5Y cell line. However, it's important to acknowledge the limitations inherent in all in vitro models when interpreting these results in the context of human PD. Additional experimental validation in more convincing models (in vivo PD model, iPSC-derived model or human sample if possible) would strengthen your conclusions. Otherwise, please address the limitation of your study clearly in the manuscript.

3. Some of the figures contain information that is unclear or illegible. Please ensure that all submitted figures meet the required resolution standards, and provide high-resolution original images for each figure.

4. Please provide a more comprehensive discussion of the study's limitations.

[Statistical Analysis]

1. The use of ROC curves and AUC for diagnostic model evaluation is appropriate. The typical way to assess a diagnostic method with ROC is to calculate the AUC for interpretation and select the optimal threshold to balance the sensitivity and specificity. Your manuscript lacks this detailed analysis. Additionally, you describe the ROC as showing "feasibility and capability" rather than the standard "sensitivity and specificity". Please revise this to use standard terminology, or provide a clear rationale for the alternative phrasing.

2. There's no discussion of potential overfitting or how it was addressed. Given the relatively small sample size, this is an important consideration.

[Data Availability]

1. You state that all data are fully available without restriction, which is commendable. However, please ensure that all data, including any code used for the bioinformatics analysis, are made available in a public repository with clear access instructions in the manuscript.

[Manuscript Presentation and Language]

The language of the manuscript appears to be generally clear and scientific. However, there are some areas where the language could be improved:

1. There are occasional grammatical errors and awkward phrasings.

2. Some sentences are long and complex, which can make them difficult to follow. For example, in the abstract:"We aimed to identify the ferroptosis-related genes (FRGs) associated with PD and construct competing endogenous RNA (ceRNA) networks by using bioinformatics analysis to further assess the pathogenesis of PD." Breaking these into shorter, more focused sentences could improve readability.

6. PLOS authors have the option to publish the peer review history of their article (what does this mean?). If published, this will include your full peer review and any attached files.

Do you want your identity to be public for this peer review? For information about this choice, including consent withdrawal, please see our Privacy Policy.

Reviewer #1: No

Reviewer #2: No

Reviewer #3: No

---

## [Author Response · Author response to Decision Letter 0]

27 Nov 2024

Dear editor’s and reviewer’s,

Thanks for your comments concerning our manuscript. They are all valuable and very important to guide our research. We have studied these comments carefully and revised the manuscript which we hope to meet with approval. Most importantly, we re-performed and added some statistical method, bioinformatic analysis especially the single-cell sequencing or experiments, which may help to fully answer the reviewer’s comments and improve the quality of the manuscript. The main revisions have been marked in the “Revised Manuscript with Track Changes”, and the “Response to Reviewers” comments are as follow. We hope our revision and the improvement of the manuscript can answer your questions.

Point-to-point response to Reviewer 1:

Question 1: The primary concern is using the sva package to remove batch effects, which is not recommended. Instead, batch should be included as a covariate in statistical models. The manuscript at https://doi.org/10.1093/biostatistics/kxv027 is highly recommended for further guidance. Programs like ComBat aim to modify the data to eliminate batch effects, but it is better to model the batch effect in the design formula without directly modifying the data.

Response: Thanks for your comment. We really appreciate your advice and it is very confirmed and valuable suggestion. It is better to include batch effects as a covariate in all statistical tests instead of removing batch effects directly with “ComBat” or “removeBatchEffect” etc. The combat modifies the expression of the entire matrix through batch effects, which sometimes may lead to negative values or overfitting. We apologize for the lack of clarity in the methods section and we modified it. Initially, limma allows batch effects, like any other factor, to be included as covariates during normalization (providing the batch information in the limma and DESeq2 packet grouping information) were performed, however, the effect is not good. Thus, we attempted to optimize the “ComBat” algorithm of the sva package. During the data preprocessing, we firstly performed background corrected on each dataset, the quality normalized were achieved after merging them. We used the model.matrix() function to construct a design model that should include the batch information as the covariates, and the Bayesian approach were used to estimate the batch effects (Setting par.prior=TRUE). We also performed Combat multiple times then choose the best result. We have added the results of the batch effect removal in supplementary figures and the boxplot indicates that the batch effect has been significantly eliminated.

Question 2: The methodology section lacks clarity and detail in certain areas. For instance, the specific methods used to correct the background and normalize the data are unclear. It is also uncertain whether any filters were applied to exclude probes with no expression or if quality control measures such as principal component analysis were conducted. 

Response: Thanks for your comment and we sincerely apologize for the lack of clarity in the methodology section. We have made detailed revisions to the 2.1 section (Data Collection and Processing). We elaborated the corresponding R packages and algorithms that used for the background correction, probes filter, datasets transformation or normalization and others in detail.

We also included the diagram of the principal component analysis in the supplementary figures.

Question 3: the authors do not appear to have addressed potential confounding factors, and it seems that multiple testing corrections were not utilized. The rationale for selecting |log2 fold change | > 0.1 as the threshold for differential expression analysis is unclear. 

Response: Thanks for your suggestion and it is a very correct and valuable suggestion. We sincerely apologize for the lack of clarity in our explanation of the DEGs analysis methods and for the limitations present in it. We used the R package “limma” for DEGs analysis. We constructed the expression matrix based on the confounding factors, used the “lmfit” function to fit a linear model, and conducted Bayesian testing. The p-value was adjusted using the BH method in order to control the FDR. However, the correction results are not very satisfactory, and the number of DEGs (adjust p<0.05) is relatively small. The reason may be related to the different batch sources of our samples after correction and the small sample size. Furthermore, in neurodegenerative diseases such as parkinson's disease, subtle changes of DEGs may lead to alterations in many related pathological pathways and molecular biology mechanisms. Therefore, we set a broader threshold for logFC or p-values, and we validated the hub genes through the experiment. We apologize for the confusion this has caused to you, we have revised the methodology section and discussed it in the limitations.

Question 4: The authors should also clarify the reasoning behind the selection of datasets, as the WGCNA analysis was carried out using only one dataset, while the DEGs analysis involved different datasets.

Response: Thanks for the invaluable comment and this is a detail worth clarifying. WGCNA was used to evaluate the relationships between hub modules and clinical traits of PD. The quality requirements for the samples are high. The four datasets are derived from sequencing conducted by different platforms or companies. We attempted to perform WGCNA on the different datasets, however, due to the large sample size and number of genes, the resulting gene modules seems that were not very satisfactory. Compared to other datasets, The GSE20292 has a better quality and the sample size meets the requirements for WGCNA, particularly with well-documented clinical information. Furthermore, there are no outlier samples or anomalous values, and the patients and control groups are matched in terms of sex and age. Therefore, we believe that conducting WGCNA on this dataset can yield modules significantly related to clinical features of PD. Thanks for your suggestions, and we have also made corresponding revisions and explanations to the 2.5 section.

Question 5: The authors are strongly encouraged to review the "Minimum Information for Publication of Quantitative Real-Time PCR Experiments" (MIQE) guidelines. The MIQE publication provides detailed information on the minimal and desirable reporting requirements for qPCR data publication.

Response: Thanks for your suggestions and they are extremely valuable. We have studied and referenced the MIQE guideline and related literatures. We have rewritten the quantitative real-time PCR methods and the details refer to the 2.13 section.

Point-to-point response to Reviewer 2:

Question 1: The manuscript would benefit from including single-cell expression analysis of NEAT1, NEUROD4, GNGT1, FCN2, and S100A2 in single-cell human post-mortem tissue (Kamath et al., Nature Neuroscience, 2022) and in single-cell midbrain organoids (Fiorenzano et al., Nature Communications, 2021). This addition would not only strengthen the authors’ findings but also highlight the potential use of brain organoids in future studies to explore the identified regulatory axis in more physiologically relevant conditions.

Response: We sincerely thanks for your suggestions and they are extremely valuable and it is very helpful for our research. Kamath et al developed a protocol to enrich and transcriptionally profile DA neurons from patients with PD and matched controls, sampling a total of 387,483 nuclei, including 22,048 DA neuron profiles (Kamath et al., Nature Neuroscience, 2022). Fiorenzano et al focused to a greater extent on the developmental processes of dopaminergic neurons (Fiorenzano et al., Nature Communications, 2021). Based on the database GSE178265 (ncbi.nlm.nih. gov/geo/query/acc.cgi?acc=GSE178265) provided by Kamath et al, we conducted further validation of the feature genes through single-cell sequencing analysis. We have added the relevant methods, results and discussion sections. We confirmed that NEAT1 is significantly upregulated in dopaminergic neurons in PD and further clarified its distribution within cellular subpopulations, which providing new directions for exploring the molecular biological mechanisms of PD. Additionally, we found that the distribution differences of some proteins within the dopaminergic neuron subpopulations are not very pronounced. Since our another study is conducting an in-depth exploration of the molecular mechanisms of one of the feature genes, which involves aspects of single-cell sequencing. We sincerely apologize for only supplementing the relevant results regarding NEAT1 here. Thank you very much for your suggestions, and we hope that our future research can also benefit from your guidance.

Question 2: The Materials and Methods section, particularly regarding the experimental use of 6-OHDA, requires further detail and clarification to improve reproducibility and transparency.

Response: Thank you for your comments and they are very crucial for our research. We sincerely apologize for the omissions in our description. We have revised the materials and methods related to 6-OHDA administration and added relevant details.

Question 3: The use of SH-SY5Y cells may not fully capture the complexity of the identified axis. It would be beneficial for the authors to show tyrosine hydroxylase (TH) expression and potential fragmentation at the immunofluorescence level before and after 6-OHDA treatment. This would provide a more comprehensive evaluation of the cellular model and the relevance of the regulatory pathway under study.

Response: We sincerely appreciate your suggestion. it is very valuable and will be useful as we further explore the molecular biology mechanisms. The reduction of TH in dopaminergic neurons is a significant feature of PD. Previous studies have demonstrated that the 6-OHDA, serves as an specific inducer for PD models, could significantly inhibit the TH expression at the immunofluorescence levels (He X, et.al.Front Pharmacol. 2021), (Guo S, et.al.Med Princ Pract. 2024). We also validated that 6-OHDA can inhibit the expression level of TH in SH-SY5Y cells through the western blot experiments. We found that compared with the control or erastin group, expression of TH was decreased in the 6-OHDA group. It indicated that the PD cell model has been successfully established under the 6-OHDA induction, which can be used for further exploration of the molecular biological mechanisms. Thank you sincerely for your reminding and attention.

Point-to-point response to Reviewer 3:

Question 1: The machine learning model validation process needs more detail. While you mention using training, test, and external validation sets, the manuscript lacks information on cross-validation procedures, which are crucial for ensuring model robustness.

Response: Thanks for your comments and the suggestion is extremely valuable and critical. We apologize for the confusion caused by our nonstandard description of machine learning. We have modified the material and methodology section related to machine learning, added to the cross-validation procedures and relevant parameters or criteria for machine learning. The details refer to the 2.7 section.

Question 2: The biological significance of the identified NEAT1/ miR-7b-5p/S100A2 axis in PD were verified in SH-SY5Y cell line. However, it's important to acknowledge the limitations inherent in all in vitro models when interpreting these results in the context of human PD. Additional experimental validation in more convincing models (in vivo PD model, iPSC-derived model or human sample if possible) would strengthen your conclusions. Otherwise, please address the limitation of your study clearly in the manuscript.

Response: Thanks for your comments and your suggestions are very valuable and helpful for our follow-up researches. We supplemented some single-cell sequencing results to support our conclusions but we apologize that in vivo and iPSC-derived model experimental verification was not performed in this study. We have explained this in the limitations section. We will further explore the molecular biological mechanism in vivo model in the future, and we also looking forward to your guidance in our future research.

Question 3: Some of the figures contain information that is unclear or illegible. Please ensure that all submitted figures meet the required resolution standards, and provide high-resolution original images for each figure.

Response: Thanks for your comments and we are very sorry for the inconvenience caused by the quality problems of the images. We have checked and uploaded higher resolution images.

Question 4: Please provide a more comprehensive discussion of the study's limitations.

Response: Thanks for your comments and they are extremely valuable. We have added the discussion of the limitations in the last paragraph.

Question 5: The use of ROC curves and AUC for diagnostic model evaluation is appropriate. The typical way to assess a diagnostic method with ROC is to calculate the AUC for interpretation and select the optimal threshold to balance the sensitivity and specificity. Your manuscript lacks this detailed analysis. Additionally, you describe the ROC as showing "feasibility and capability" rather than the standard "sensitivity and specificity". Please revise this to use standard terminology, or provide a clear rationale for the alternative phrasing.

Response: Thanks for your suggestions and we sincerely apologize for the confusion caused by our non-standard writing. The suggestion is extremely valuable for our research. We have made corresponding revisions to the methods and results sections related to ROC, especially in terms of wording and the description of additional details.

Question 6: There's no discussion of potential overfitting or how it was addressed. Given the relatively small sample size, this is an important consideration.

Response: Thanks for your valid suggestion that is very valuable and crucial for our research. The sample size of this study is relatively small, so we used 10-fold or 5-fold cross-validation and regularization techniques to limit the model weights in machine learning to reduce overfitting. We have provided additional explanations in the methodology section related to machine learning.

Question 7: You state that all data are fully available without restriction, which is commendable. However, please ensure that all data, including any code used for the bioinformatics analysis, are made available in a public repository with clear access instructions in the manuscript.

Response: Thanks for your comments and that is very important suggestion. We have appended a description that provide the public website of all the databases, genesets or the associated codes. The details are described in the statement of the data availability section. Thank you sincerely for your reminding and attention.

Question 8: There are occasional grammatical errors and awkward phrasings.

Response: We sincerely appreciate your suggestions and apologize for the confusion caused by the grammatical errors in our writing. We have reviewed the entire manuscript and made revisions to the wording and polished the relevant statements.

Question 9: Some sentences are long and complex, which can make them difficult to follow. For example, in the abstract:"We aimed to identify the ferroptosis-related genes (FRGs) associated with PD and construct competing endogenous RNA (ceRNA) networks by using bioinformatics analysis to further assess the pathogenesis of PD." Breaking these into shorter, more focused sentences could improve readability.

Response: Thank you for your suggestions and they are very valuable. We have revised and simplified some lengthy sentences in the the abstract and main text.

Point-to-point response to Reviewer 4:

Question 1: The first author, Dr. Li is also the second author of a similar paper titled "Identifying the potential genes in alpha synuclein driving ferroptosis of Parkinson’s disease" published last year. It is not so surprising that the conclusion is different between the past paper and the current manuscript since a distinct n

---

## [Decision Letter · Decision Letter 1]

9 Dec 2024

Identifying the NEAT1/miR-26b-5p/S100A2 Axis as a Regulator in Parkinson’s Disease based on the Ferroptosis-Related Genes

PONE-D-24-16545R1

Dear Dr. Guo,

We’re pleased to inform you that your manuscript has been judged scientifically suitable for publication and will be formally accepted for publication once it meets all outstanding technical requirements.

Kind regards,

Alexis G. Murillo Carrasco

Academic Editor

PLOS ONE

Additional Editor Comments (optional):

Reviewers' comments:

Reviewer's Responses to Questions

**Comments to the Author**

1. If the authors have adequately addressed your comments raised in a previous round of review and you feel that this manuscript is now acceptable for publication, you may indicate that here to bypass the “Comments to the Author” section, enter your conflict of interest statement in the “Confidential to Editor” section, and submit your "Accept" recommendation.

Reviewer #1: All comments have been addressed

Reviewer #2: (No Response)

Reviewer #3: All comments have been addressed

2. Is the manuscript technically sound, and do the data support the conclusions?

Reviewer #1: Yes

Reviewer #2: Yes

Reviewer #3: Yes

3. Has the statistical analysis been performed appropriately and rigorously? 

Reviewer #1: Yes

Reviewer #2: Yes

Reviewer #3: Yes

4. Have the authors made all data underlying the findings in their manuscript fully available?

Reviewer #1: Yes

Reviewer #2: Yes

Reviewer #3: Yes

5. Is the manuscript presented in an intelligible fashion and written in standard English?

Reviewer #1: Yes

Reviewer #2: Yes

Reviewer #3: Yes

6. Review Comments to the Author

Reviewer #1: (No Response)

Reviewer #2: The reviewer thanks the authors for addressing the earlier comment. However, upon reviewing the revised manuscript, it remains unclear how the datasets from Kamath et al. (Nature Neuroscience, 2022) and Fiorenzano et al. (Nature Communications, 2021) have been incorporated or compared with the authors’ findings. Specifically, it is not evident from the text or figures where this comparison is presented.

The reviewer kindly requests that the authors clearly indicate the relevant sections (e.g., page numbers) and figures where this integration or comparison is addressed. If this has not yet been included, the reviewer encourages the authors to provide explicit details or analyses, as previously suggested, to ensure that the potential relevance of these datasets to the study’s findings is thoroughly explored and appropriately highlighted.

This clarification would significantly improve the clarity of the manuscript and its contribution to the field.

Reviewer #3: Overall, the revised manuscript is suitable for consideration in the publication process. The authors have made significant improvements to the quality of their manuscript and provided appropriate responses to the reviewers' concerns. For aspects that could not be fully addressed, they provided reasonable explanations and suitable alternatives.

The statistical analysis is appropriate. The authors have applied suitable statistical methods, addressed previous concerns about methodology and overfitting, and enhanced the manuscript's clarity and transparency. The analysis supports the study's conclusions.

7. PLOS authors have the option to publish the peer review history of their article (what does this mean?). If published, this will include your full peer review and any attached files.

Reviewer #1: No

Reviewer #2: No

Reviewer #3: No

---

## [Editor Report · Acceptance letter]

20 Dec 2024

PONE-D-24-16545R1 

PLOS ONE

Dear Dr. Guo, 

I'm pleased to inform you that your manuscript has been deemed suitable for publication in PLOS ONE. Congratulations! Your manuscript is now being handed over to our production team.

Kind regards, 

on behalf of

Dr. Alexis G. Murillo Carrasco 

Academic Editor

PLOS ONE